# Life at high temperature observed in vitro upon laser heating of gold nanoparticles

Céline Molinaro [1,4], Maëlle Bénéfice[1,4], Aurore Gorlas[2], Violette Da Cunha[2], Hadrien M. L. Robert [1], Ryan Catchpole[2], Laurent Gallais[1], Patrick Forterre[2,3] & Guillaume Baffou [1] ✉

Thermophiles are microorganisms that thrive at high temperature. Studying them can provide valuable information on how life has adapted to extreme conditions. However, high temperature conditions are difficult to achieve on conventional optical microscopes. Some home-made solutions have been proposed, all based on local resistive electric heating, but no simple commercial solution exists. In this article, we introduce the concept of microscale laser heating over the field of view of a microscope to achieve high temperature for the study of thermophiles, while maintaining the user environment in soft conditions. Microscale heating with moderate laser intensities is achieved using a substrate covered with gold nanoparticles, as biocompatible, efficient light absorbers. The influences of possible microscale fluid convection, cell confinement and centrifugal thermophoretic motion are discussed. The method is demonstrated with two species: (i) *Geobacillus stearothermophilus*, a motile thermophilic bacterium thriving around 65 °C, which we observed to germinate, grow and swim upon microscale heating and (ii) *Sulfolobus shibatae*, a hyperthermophilic archaeon living at the optimal temperature of 80 °C. This work opens the path toward simple and safe observation of thermophilic microorganisms using current and accessible microscopy tools.

Over billions of years, life on Earth has evolved to adapt to a large variety of environmental conditions, sometimes considered extreme from our human perspective. In particular, some heat-loving microorganisms (bacteria, archaea, fungi), called thermophiles, thrive at temperatures ranging from 45 °C to 122 °C[1–4]. Thermophiles live in various ecological systems like deep-sea hydrothermal vents, hot springs or volcanic sites. Their study sparked much interest the past decades for at least two reasons. First, we can learn from them, e.g., how thermophilic proteins[5,6], enzymes[7,8] and membranes[9] can be stable at such high temperatures, or how thermophiles manage to withstand extreme radiation levels[10]. Second, they are at the basis of many important biotechnological applications[1,11,12], such as fuel generation[13–16], chemical synthesis (dihydrogen, alcohols, methane,

amino-acids, etc)[17], biomining[18], and generation of thermostable biocatalysts[7,11,13]. In particular, the now-famous polymerase chain reaction (PCR)[19] involves an enzyme (Taq polymerase) isolated from the thermophilic bacterium *Thermus aquaticus*, one of the earliest discovered thermophiles.

However, the study of thermophiles is not straightforward, and cannot be improvised in any biolaboratory. In particular, the observation of living thermophiles in vitro cannot be conducted on any standard optical microscopes, even equipped with heating chambers available on the market, which are usually not designed to exceed 40 °C. Since the 90 s, only a few research groups have worked on the implementation of high-temperature microscopy (HTM) systems. In 1994, Gluch et al. conceived a heating/cooling chamber based on the

[1]Institut Fresnel, CNRS, Aix Marseille University, Centrale Marseille, Marseille, France. [2]Université Paris-Saclay, CEA, CNRS, Institute for Integrative Biology of the Cell (I2BC), 91198 Gif-sur-Yvette, France. [3]Département de Microbiologie, Institut Pasteur, 25 rue du Docteur Roux, 75015 Paris, France. [4]These authors contributed equally: Céline Molinaro, Maëlle Bénéfice. ✉e-mail: guillaume.baffou@fresnel.fr

use of Peltier cells controlling the temperature of a rectangular capillary that was sealed to maintain anaerobicity[20]. This device could heat up to 100 °C at a rate of 2 °C/s and enabled the authors to investigate the motility of the hyperthermophilic bacterium *Thermotoga maritima*[21]. In 1999, Horn et al. developed a very similar device, still based on the use of a heated capillary adapted on a commercial microscope, to investigate cell division/interconnection[22]. After a long period of relative inactivity, the search for efficient HTMs has gained a renewed interest from 2012, in particular with a series of articles from the Wirth group, who used the device invented by Horn et al. 15 years earlier, based on the use of heated capillaries, to investigate the motility of a large variety of archaea, including hyperthermophiles up to 100 °C[23,24]. They also upgraded the original microscope to enable faster heating (a few minutes instead of 35 min to reach the targeted temperature) and to achieve a linear temperature gradient throughout the medium, over 2 cm. This temperature gradient-forming device (TGFD) was used to study the motility of many thermophiles within temperature gradients over biotope-relevant distances[24,25].

The heating of sealed capillaries was not the only approach developed to observe living thermophiles. In 2012, Kuwabara et al. used hand-made, disposable Pyrex chambers, sealed with high-temperature durable glue (Super X2; Cemedine, Japan). The sample was placed onto a commercial, transparent hot plate (Micro Heat Plate, Kitazato Corporation, Japan) capable of heating up to 110 °C but that was not originally designed for bioimaging[26]. The authors observed the effective division of an anaerobic thermophilic bacterium at 65 °C (*Thermosipho globiformans*, with doubling time of 24 min). In 2020, Pulschen et al. demonstrated the efficient heating of commercial metallic dishes (Attofluor™, Thermofisher) using two home-made heating elements: a cap and a stage (a configuration inspired by PCR machines). This association led to uniform temperature in the liquid, and avoided evaporation and condensation on the underside of the lid. Any gas exchange with the surroundings was avoided using O-rings. This HTM (called the Sulfoscope) was used to image *Sulfolobus acidocaldarius* at 75 °C[27].

A recognized limitation of all these systems was the restriction to the use of air objectives, any oil immersion being unsuited for such high temperature and for imaging through >1-mm thick transparent samples. Very recently, this limitation was lifted by Charles-Orszag et al.[28] who developed a device where heating was no longer supplied to the surroundings of the system of interest, but within the glass coverslip itself, which was covered with a thin transparent and resistive layer made of ITO (indium tin oxide). By passing a current through the transparent layer, the cover could be heated up to 75 °C. However, the authors also had to heat the objective, but not to more than 65 °C to avoid damaging it.

These works show that the development of effective high-temperature light microscopy is not widespread, usually requires home-made devices, and is often achieved at the expense of spatial resolution, a major drawback considering that thermophilic microorganisms are a few microns in size at most. Reducing the heated volume is key to solving the three inherent problems of HTMs: poor spatial resolution, large thermal inertia of the system upon heating, and the deleterious heating of surrounding elements at extreme temperature (the immersion oil, the objective… or the user's hand).

In this article, we introduce a HTM for the observation of thermophiles that is not based on resistive heating. Instead, we achieve local heating over a restricted area of the microscope's field of view by laser illumination of a light-absorbing substrate. The temperature distribution is imaged by quantitative phase microscopy (QPM). The effectiveness of the technique is illustrated with *Geobacillus stearothermophilus*, a motile thermophilic bacterium thriving around 65 °C with a short doubling time (around 20 min), and on *Sulfolobus shibatae*, a hyperthermophilic archaeon growing optimally at 80 °C. Normal replication rates and swimming were observed as a function of the

temperature. This laser-assisted HTM (LA-HTM) does not suffer from limitations regarding the coverslip thickness, nor the nature of the objective (air or oil immersion). It allows the use of any high-resolution objective on the market. It also does not suffer from the drawbacks of slow heating due to thermal inertia (instantaneous heating on the millisecond time scale is achieved) and it only involves commercially available components. The only new safety issue stems from the presence of a high-power laser beam (up to typically 100 mW) within the setup and possibly throughout the ocular, which requires the wearing of safety glasses.

## Results

### Principle of laser-assisted high-temperature microscopy (LA-HTM)

The principle of the LA-HTM consists in locally heating a sample within the field of view of a microscope using a laser (Fig. 1a). For this purpose, the sample has to be light-absorbing. In order to use reasonable laser powers (less than 100 mW), we do not rely upon light absorption of the liquid medium, but rather artificially increase the sample absorbance by covering the substrate with gold nanoparticles (Fig. 1c). Heating gold nanoparticles with light is at the basis of the field of thermoplasmonics, with envisioned applications in biomedicine, nanochemistry or solar light harvesting[29–31]. We have been using this LA-HTM in several studies these last years, related to thermoplasmonics applications in physics, chemistry and biology. The main difficulty in this approach is to map the resulting temperature distribution, because the elevated temperatures are confined to microscale regions within the sample. We have shown that temperature mapping can be achieved using quadriwave lateral shearing interferometry[32], a simple, high-resolution, and very sensitive quantitative phase microscopy technique based on the use of a 2-dimensional diffraction grating (aka cross-grating)[33–36]. The reliability of this temperature microscopy technique based on cross-grating wavefront microscopy (CGM) has been proven though the publication of a dozen of articles this last decade[37–43].

In particular, we recently achieved mammalian cell heating using the LA-HTM and CGM, and followed the cellular heat-shock response in the range 37–42 °C, demonstrating the applicability of the technique for live-cell imaging at the single-cell level[41]. However, the application of the LA-HTM to study microorganisms at high temperature is not straightforward, as it requires more caution compared with mammalian cells: first, heating the bottom of the medium by a few tens of degrees (rather than a few degrees) creates a strong vertical temperature gradient within the fluid likely to yield fluid convection[44], inducing undesired motion and mixing of the bacteria if they are not firmly anchored to the substrate. Such convection can be abrogated by reducing the thickness of the liquid layer. For this purpose, in all the experiments we present hereinafter, the bacterial suspensions were sandwiched between two coverslips at a thickness of around 15 μm, mounted within a metallic dish (Attofluor™, Thermofisher, Fig. 1b, c). In principle, convection can be avoided if the liquid thickness is less than the beam size of the heating laser[44]. Second, working in such a confined geometry may asphyxiate aerobic organisms (see Fig. S2). This problem can be avoided by using a substrate permeable to oxygen (or to any other vital gas), leaving a trapped air bubble within the coverslips or by drilling a hole within the top coverslip (see Fig. S1)[45]. In this study, we opted for the latter solution (Fig. 1b and S1). Finally, heating with a laser does not yield a uniform temperature profile. Even if using a uniform laser beam intensity (Fig. 1d), the temperature distribution is not uniform, rather resembling a Gaussian distribution due to thermal diffusion (Fig. 1e). A non-uniform profile is not ideal when the aim is to set a precise temperature over the field of view for the study of a biological system, and may also cause thermophoretic motion of the bacteria if they are not adherent to the substrate (see Figs. S3, S4)[39]. For this purpose, we used a spatial light modulator (SLM) to shape the infrared laser beam at the sample plane according

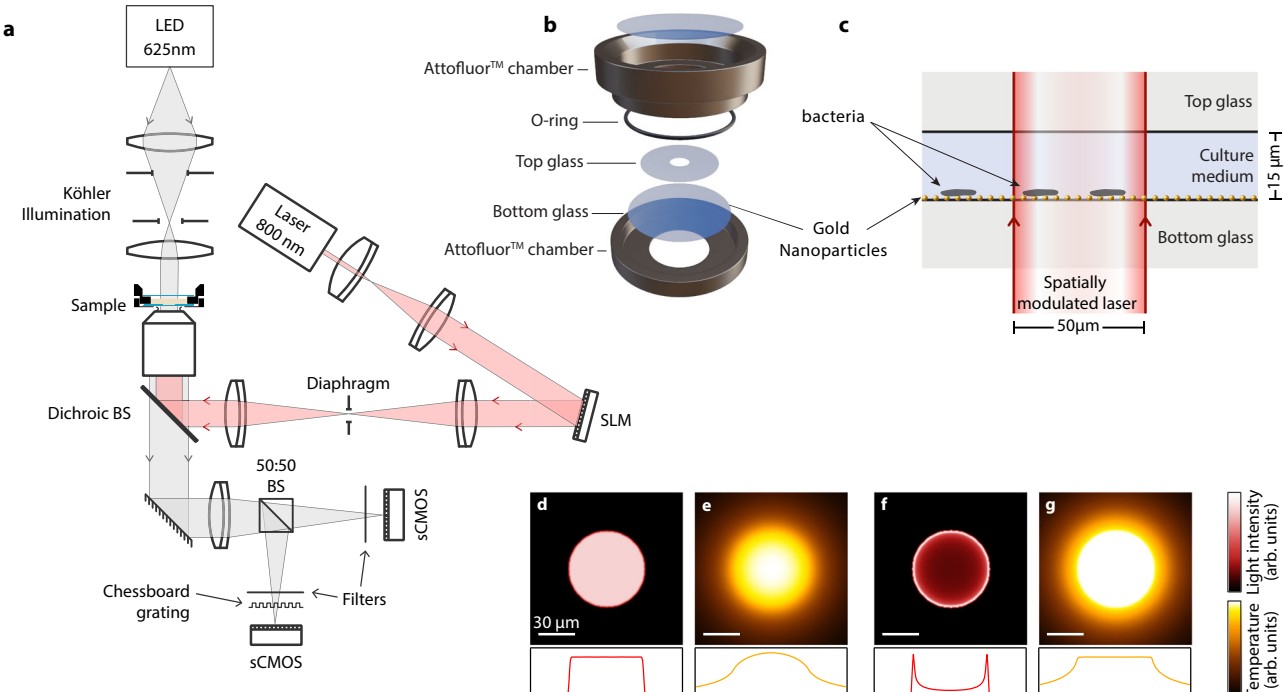

**Fig. 1 | Experimental approach. a** Schematic of the setup for parallel laser heating, shaping and temperature microscopy. **b** Sample geometry, consisting of an Attofluor™ chamber containing a coverslip covered with gold nanoparticles. **c** Close look at the sample (not to scale). **d** Representation of a uniform laser beam profile and (**e**) simulation of the subsequent temperature distribution at the gold nano-particle sample plane. **f** Representation of a ring-like laser beam profile suited for uniform temperature generation, as shown in the simulation of the resulting temperature distribution displayed in (**g**). Scale bars: 30 μm.

to a ring shape (Fig. 1f) to achieve a perfectly uniform temperature distribution over a given geometric area, despite thermal diffusion (Fig. 1g)[39,42,46]. A top coverslip is put on top of the metallic dish (Fig. 1b) to avoid evaporation of the medium and enable observation of at least a couple of days. Because this top coverslip is not sealed, additional medium can be easily added at any time, if need be.

## Laser activation of *G. stearothermophilus* growth

To illustrate the working principle of the LA-HTM, and demonstrate its applicability to the study of thermophiles, we studied *Geobacillus stearothermophilus*, an aerobic bacterium capable of fast growth (doubling time of approximately 25 min), with an optimal growth temperature around 60–65 °C. This bacterium also possesses a fla-gellum and swimming capabilities, offering another readout of normal cell activity.

The sample (Fig. 1b) was pre-incubated for one hour at 60 °C before being placed on the LA-HTM sample holder. This pre-incubation is not a requirement, but remains useful for two reasons: First, it leads to an immediate onset of cell growth and division when the laser is turned on (see Movie M1 in Supplementary Materials). Without pre-incubation, bacterial growth tends to be delayed by around 40 min each time a new observation area is heated on the sample. Second, the 1-h long pre-incubation favors the adhesion of bacteria onto the coverslip, which prevents cells from drifting out of the field of view due to thermophoresis when the laser is turned on (see Movie M2 in Supplementary Materials). Thermophoresis denotes the motion of particles or molecules along temperature gradient, usually from hot to cold, and bacteria are not an exception[43,47]. This undesired effect is cancelled over a given area when using an SLM to shape the laser beam and achieve a flat temperature profile.

Figure 2 shows the temperature distribution measured using CGM, obtained when illuminating the glass substrate covered with gold nanoparticles with a ring-like laser beam (Fig. 1f). A flat temperature profile is observed throughout the area covered by the laser

beam. This area was set at 65 °C, the optimal growth temperature. Outside this area, the temperature profile naturally decays as $1/r$ (where $r$ is the radial coordinate).

The activity of bacterial cells was monitored over several hours with the LA-HTM. Figure 3 presents a time lapse of four images taken from a single 3h20′ long movie (Movie M3, Supplementary Information). Bacteria were observed to actively grow within the circular area defined by the laser, where the temperature is optimal, close to 65 °C. In contrast, 10 s of micrometers away where the temperature drops below 50 °C, cell growth was significantly reduced.

## LA-HTM for easy analysis of growth temperature of microorganisms

To further quantify cell growth and its dependence on temperature, we measured the increase in biomass of the different colonies stem-ming from initial isolated bacteria lying within the field of view of Movie M3 (Fig. 4). The selected initial bacteria at the origin of the microscale colony forming unit (mCFU) are indicated in Fig. S6. Dry mass measurements were performed with the CGM camera[48], the same camera that is used to map the temperature distribution. The ability of CGM to measure both dry mass and temperature is a strength of the LA-HTM. As expected, high temperatures yield faster bacterial growth (Fig. 4a). Growth at all temperatures follows an exponential increase, as demonstrated by the semi-log plot of Fig. 4b, where the data were fitted using an exponential function $m = m_0 10^{t/\tau} + cst$, where $\tau\log2$ is the generation time (or doubling time) and $g = 1/\tau$ is the growth rate (number of division per unit time). Figure 4c plots the corresponding growth rate and generation time as a function of temperature. The fast growing mCFUs feature a saturation of growth after two hours, an expected behavior certainly due to the high bac-terial density (similar to stationary phase in classic liquid culture). The overall $g(T)$ shape (Fig. 4c) corresponds to the expected biphasic profile of *G. stearothermophilus*, with an optimal growth rate around 60–65 °C. Data were fitted using the Cardinal Model (Fig. S5)[49], with

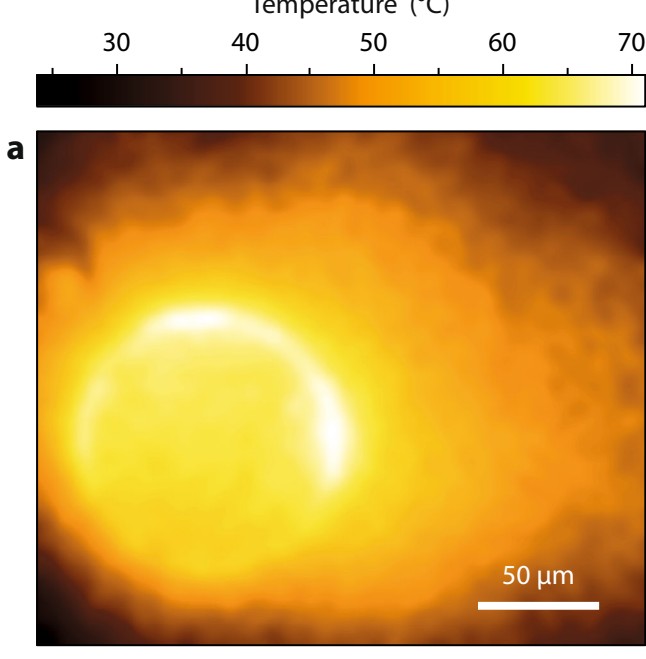

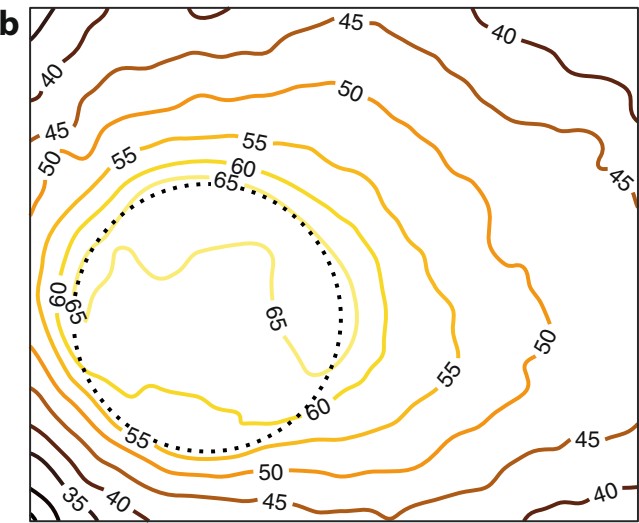

Fig. 2 | **Temperature shaping at the microscale. a** Temperature map measured using CGM obtained when illuminating a gold nanoparticle layer with a ring-like laser beam to obtain a flat temperature profile over a circular area. **b** Isotherms of the temperature map (**a**). The outline of the laser beam is represented by a grey, dotted circle. Experiment repeated twice (see Supplementary Materials, Fig. S4).

$\left(G_0; T_{min}; T_{opt}; T_{max}\right) = (0.70 \pm 0.2; 40 \pm 4; 65 \pm 1.6; 67 \pm 3)$ °C, in good agreement with other reported values from the literature[49]. While the temperature-related parameters were reproducible, the maximum growth $G_0$ rate could vary from one experiment to another (see Fig. S7–S9 and Movie M4). Unlike the temperature fitting parameters that should be universal, the maximum growth rate is dependent on the nature of the culture medium within the microscale geometry being observed (nutrient availability, oxygen concentration).

## Observation of cell motility
In addition to normal growth, some bacteria have been observed to occasionally swim throughout the field of view during laser heating, a behavior expected from bacteria possessing flagella. Movie M5 in Supplementary Information features such swimming events. In this experiment, a temperature gradient was produced using a uniform

laser illumination such as in Fig. 1d, e and S3. Figure 5 displays two sequences of images selected from Movie M5 that show single bacteria exhibiting a directional motion while all the others remain still.

In the case of *G. stearothermophilus*, the onset of active bacterial motion (Fig. 5) occurs a few seconds after the laser beam turned on. This observation highlights the short response of this thermophilic microorganism following a temperature increase, as already observed by Mora et al.[24]. Further investigations could be conducted on this topic of bacterial motility or even thermotaxis using the LA-HTM.

Microbial swimming should not be mixed with other types of physical motion, namely (i) Brownian motion, which looks like a chaotic motion with no specific orientation, (ii) convection[50] and thermophoresis[43], which consist of a regular drift motion along a temperature gradient.

## Laser activation of *G. stearothermophilus* spore germination
*G. stearothermophilus* is famous for its ability to form highly resistant spores (sporulation) when subjected to unfavourable environmental conditions as a means of protection. When external conditions become favourable again, spores germinate, producing viable cells and restoring growth. Although well-known, this sporulation/germination process has never been observed in real-time. Using the LA-HTM, we report here the first observation of germination events of *G. stearothermophilus*.

Figure 6a displays a time lapse of optical thickness (OT) images acquired using CGM of a group of 13 spores. Over the whole acquisition time (15 h 6 min, $t = 0$ is the onset of laser heating), 4 spores germinated among the 13, doing so independently, at successive times $t = 2$ h, 3 h 10', 9 h 40' and 11 h 30'. While Fig. 6 displays only 1 of these events, the 4 germination events can be observed in Movie M6 in Supplementary Materials. Interestingly, germination looks stochastic: not all the spores germinate, and they do not germinate at the same time, although they experienced the exact same variations of environmental conditions.

Figure 6b, c plot the biomass of the cell population within the field of view as a function of time for the whole acquisition period. The rapid decay of the dry mass observed at $t = 5$ h in Fig. 6b, c is due to the departure of some cells from the field of view. The four events exhibit a growth rate of $0.77 \pm 0.1$ h$^{-1}$. This value is higher than the growth rate related to Figs. 3 and 4, where the cells are normally growing. The reason for this enhanced growth rate of *G. stearothermophilus* from spores is not known, but these measurements highlight the interest of the LA-HTM and of working at the single-cell level (or single mCFU level) to learn more about cell living dynamics.

## Laser activation of *S. shibatae* growth
To further demonstrate the versatility of the LA-HTM and its operation at high temperature, we studied the growth of *Sulfolobus shibatae*, a hyperthermophilic, acidophilic archaeon with an optimal growth temperature of 80 °C[51]. This archaeon also features a much different morphology compared to *G. stearothermophilus*, looking like 1-μm spheres (cocci) rather than elongated rods (bacilli).

Figure 7a consists of successive optical thickness images of a *S. shibatae* mCFU, acquired using CGM (see full-length Movie M7 in Supplementary Materials). This mCFU was growing around 73 °C, a temperature lower than the optimal 80 °C, but within the temperature range of active growth. We observe the multiple division events making the mCFU looking like a microscale grape of archaea after a few hours. From these OT images, the biomass of the mCFU was measured over time, and plotted in Fig. 7b. Interestingly, the *S. shibatae* mCFU exhibits linear growth, rather than the exponential growth observed for *G. stearothermophilus* mCFUs. There exists a long-standing debate about cellular growth rate patterns[52]: while some studies report a growth rate of microorganisms that is proportional to their size (exponential growth), some other studies

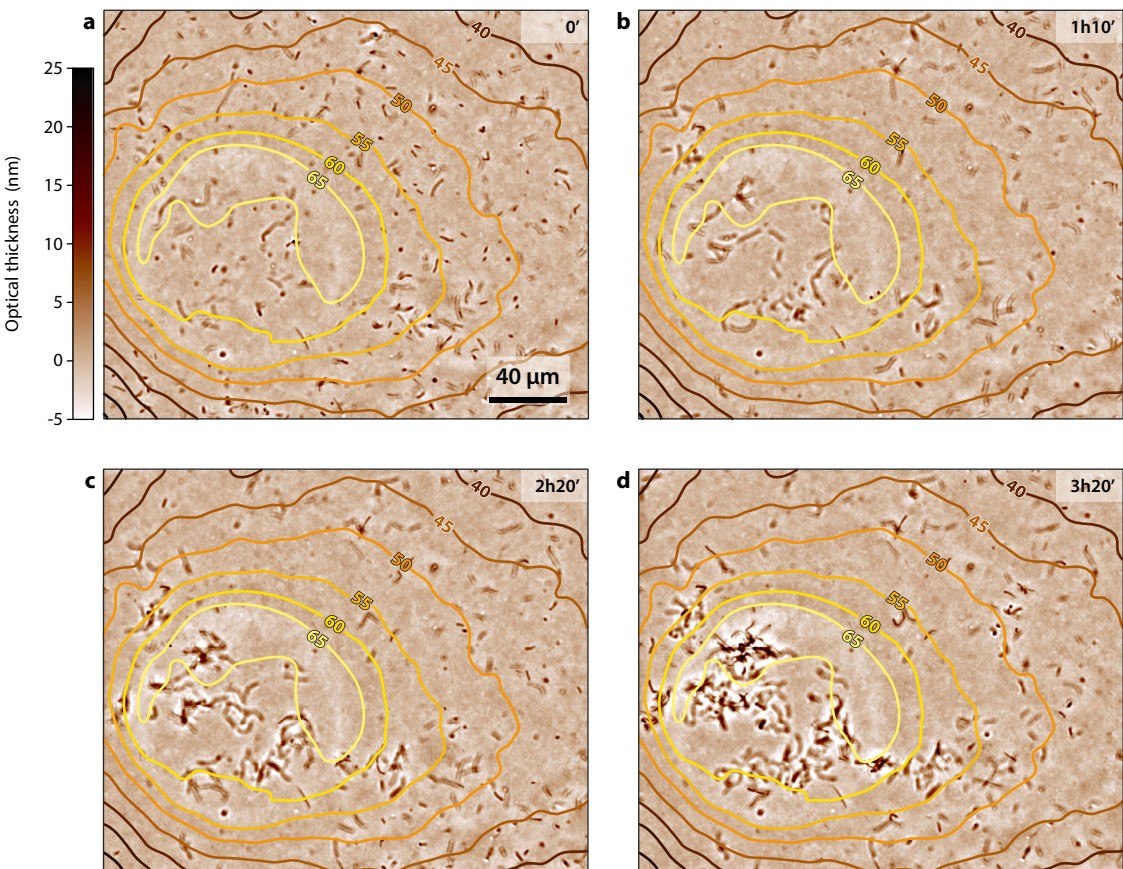

**Fig. 3 | Laser activated growth of bacteria.** Optical thickness images of the growth of *G. stearothermophilus* bacteria upon laser heating at different time, (**a**) $t = 0$ min, (**b**) 1 h 10 min, (**c**) 2 h 20 min, (**d**) 3 h 20 min, extracted from a 200 min long movie (movie M3, provided in Supplementary Information), superimposed with the associated temperature map. The laser was switched on at time $t = 0$. Isotherms have been added to the intensity images.

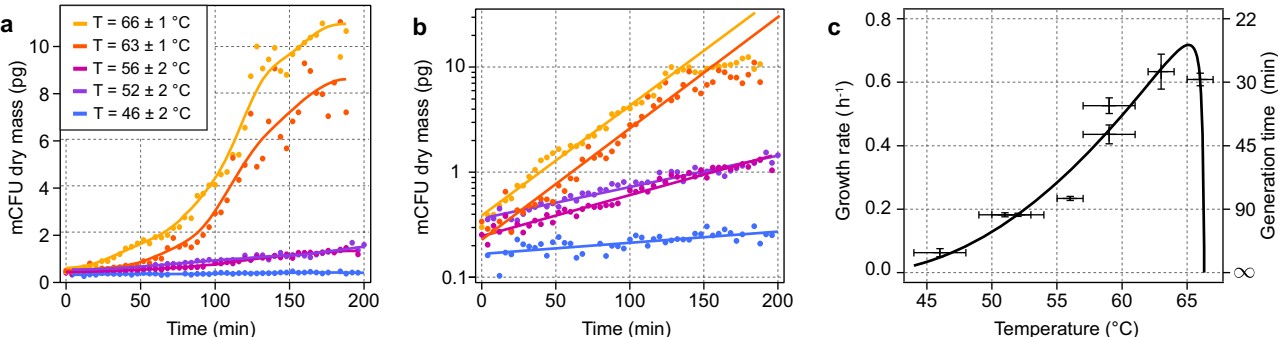

**Fig. 4 | Growth rate measurements as a function of temperature. a** Microbial growth at different temperatures. mCFU: microscale colony forming unit. Data acquired from a single movie of bacteria growing within a temperature gradient (movie M3). **b** Same as (**a**) in a semi-log scale. **c** Growth rate $\tau$ and generation time $g$ calculated from the linear regressions of (**b**). Horizontal error bar: temperature range over which the mCFU expanded over the field of view during growth. Vertical error bar: root mean square errors of the linear regressions.

rather demonstrate a constant rate (linear or bilinear growth). As explained by Tzur et al.[53], distinguishing between exponential and (bi)linear growth requires a precision of <6% in biomass measurements, which is out of reach for most QPM techniques, even involving interferometry. CGM reaches this degree of precision with sub-pg accuracy in biomass measurements[36,48].

The perfectly linear growth of *S. shibatae* was unexpected and has never been reported to date. Yet, an exponential growth was expected, at least because of the multiple divisions that should yield 2, 4, 8, 16,… cells over time. We hypothesize that the linear growth could be due to the close packing of the cells, leading to cell inhibition, for the same reason that cell growth rate slows down and eventually reaches a stationary state when the cell density is too high.

## Discussion

As a conclusion, we successively discuss below five points that deserve attention: the reduction of the heated volume, the reduction of the thermal inertia, the interest of gold nanoparticles, the interest of quantitative phase microscopy, and the temperature range the LA-HTM can potentially reach.

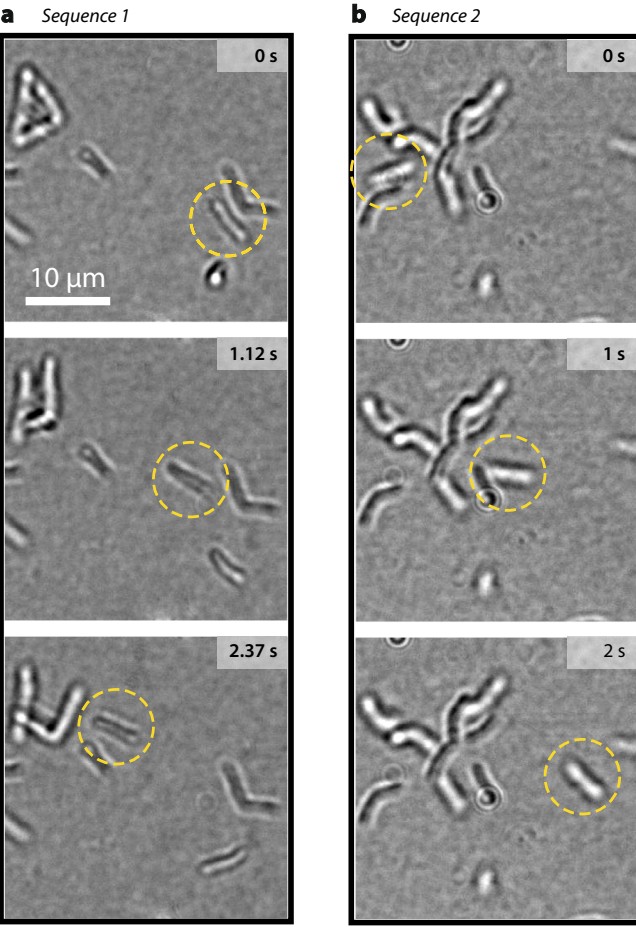

**a** *Sequence 1*

0 s

10 µm

1.12 s

2.37 s

**b** *Sequence 2*

0 s

1 s

2 s

**Fig. 5 | Bacterial swim.** Two time lapses, (**a**) and (**b**), showing the swimming of two different bacteria, spotted using dashed circles. Images were extracted from Movie M5 (available as Supplementary Materials).

Laser-heating, compared with resistive-heating, for the development of an HTM offers several benefits that we illustrate in this study. In particular, the heated volume remains confined to a volume of several $(10\,\mu m)^3$, within the liquid medium, in the field of view of the microscope. This way, only the microorganisms under observation are active, with other bacteria remaining dormant and available for further studies on the sample – there is no need to change sample each time a new temperature needs to be tested. Additionally, heating on the microscale enables the direct investigation of a wide range of temperature: Fig. 4c was obtained from a single 3-h long movie (Movie M3), which would normally require the preparation and the study of several samples – one for each temperature investigated, representing days of experiments. The reduction of the heated volume also keeps all the surrounding optical components of the microscope at room temperature, in particular the objective lens, which was a major problem encountered so far by the community. The LA-HTM enables the use of any objective lens, including oil immersion, which will remain at room temperature, even with extreme temperatures reached over the field of view. The main limitation of the laser-heating approach we report in this study is that cells that don't adhere, or that swim, may move away from the field of view and be difficult to investigate. A workaround could be to use low-magnification objectives to achieve a much more extended temperature increase, over a few hundreds of microns. This caution would come along with a reduced spatial resolution but if the aim is to study the motion of microorganisms, a high spatial resolution is not a requirement.

The time scale $\tau_D$ of heating (and cooling) of a system depends on its size, according to the law $\tau_D = L^2/D$, where $L$ is the characteristic dimension of the heat source (the laser beam diameter in our study, $L \approx 100\ \mu m$) and $D$ is the thermal diffusivity of the surroundings (the average of glass and water diffusivities in our case, $D \approx 2 \times 10^{-7}\ m^2/s$). Thus, in the presented study, we can expect a time response on the order of 50 ms, i.e., quasi-instantaneous temperature variations. This instantaneous establishment of the temperature increase not only reduces the experiment duration, but also enables the precise definition of the time $t = 0$ for any dynamical study of the effect of temperature.

The approach we propose here would work with any light-absorbing substrate (for instance commercial ITO-covered samples). However, gold nanoparticles enable strong absorption in the infrared range and reduced absorption in the visible range, the latter feature being interesting for the efficient optical observation in the visible range, in particular when using fluorescence[41]. Also, gold is bio-compatible, chemically inert, the absorbance can be tuned from 530 nm to the near-infrared, and the sample fabrication is simple and cost-effective[29].

Cross-grating wavefront microscopy (CGM) enabled not only microscale temperature mapping, but also biomass monitoring, making it particularly useful, if not necessary, in association with the LA-HTM. Other temperature microscopy techniques have been developed this last decade, especially in the field of bioimaging, and most require the use of temperature sensitive fluorescent probes[54,55]. However, such techniques have received criticism with some works reporting measurement of unrealistic temperature variations within cells, likely due to the fact that fluorescence is dependent upon many factors other than temperature[56–59]. Moreover, most fluorescent probes would not be stable at high temperature. Thus, QPMs, and in particular CGM, represents a perfect temperature microscopy technique for the study of life at high temperatures using optical microscopes.

The study of *S. shibatae*, optimally living at 80 °C, shows that the LA-HTM can be applied to the study of *hyper*thermophiles, not only to simple thermophiles. In principle, there exists no limit in the temperature range that can be achieved using the LA-HTM and temperatures higher than 100 °C could even be achieved at ambient pressure without boiling, as demonstrated by our group[38] for applications in hydrothermal chemistry at ambient pressure using laser heating of gold nanoparticles[40]. Thus, the LA-HTM could potentially be used to observe unprecedented hyperthermophiles with standard high-resolution optical microscopes, in standard conditions (i.e. ambient pressure).

## Methods
### Microscope
A home-made microscope was used to conduct all the experiments, including a Köhler illumination (with a light emitting diode, M625L3, Thorlabs, 700 mW), a sample holder with *x-y* manual translation, an objective lens (Olympus, 60x, 0.7 NA, air, LUC-PlanFLN60X, or 60x, 1.25 NA, oil, UPLFLN60XOI), a CGM camera (QLSI cross-grating, pitch of 39 µm, positioned at 0.87 mm from the sensor of a Zyla camera from Andor) allowing intensity and wavefront imaging, and a sCMOS camera (ORCA Flash 4.0 V3, 16-bit mode, from Hamamatsu) to record the data presented in Fig. 5 (bacteria swimming). The dichroic beam splitter was a 749 nm edge BrightLine (Semrock, FF749-SDi01). The filter in front of the camera was a 694 short-pass filter (FF02-694/SP-25, Semrock). A Titanium:Sapphire laser (laser Verdi G10, 532 nm, 10 W, pumping a Tsunami laser cavity, Spectra-Physics for Figs. 2–5, further replaced by a laser Millenia, Spectraphysics 10 W, pumping a Mira laser cavity, Coherent, for Figs. 6 and 7) set at a wavelength $\lambda = 800$ nm, matching the plasmonic resonance spectrum of the gold nanoparticles. The spatial light modulator (1920 × 1152 pixels) was

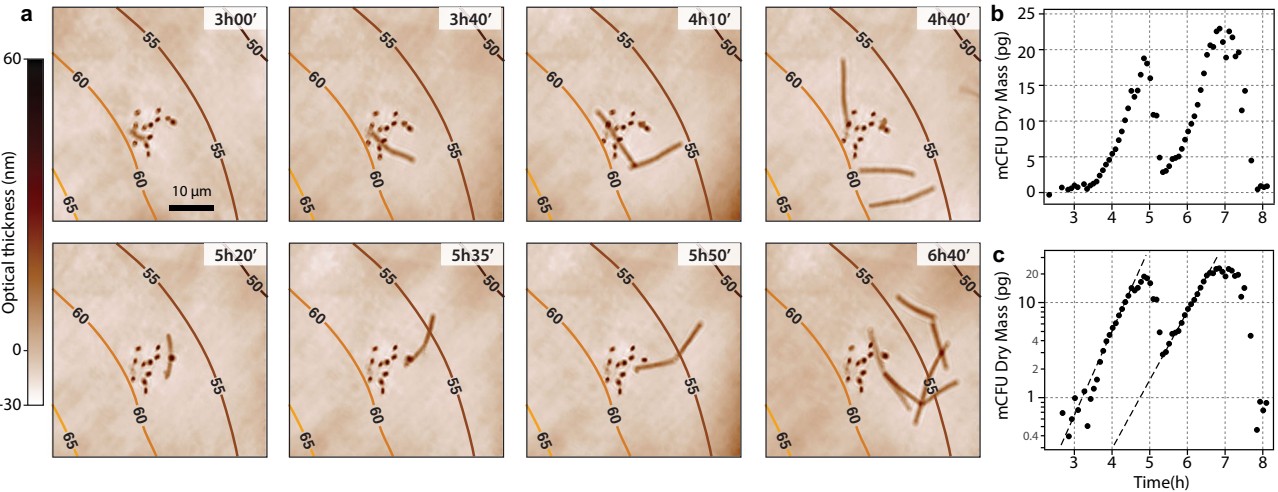

**Fig. 6 | Germination of *G. stearothermophilus* followed by CGM, and activated by laser-heating. a** Time lapse composed of 8 OT images (immersion oil, 60x, 1.25 NA objective lens), along with (**b**) the evolution of the biomass of the *G.* *stearothermophilus* aggregate. **c** Plot of (**b**) in semi-log scale to highlight the linear fits giving the growth rates (dashed lines).

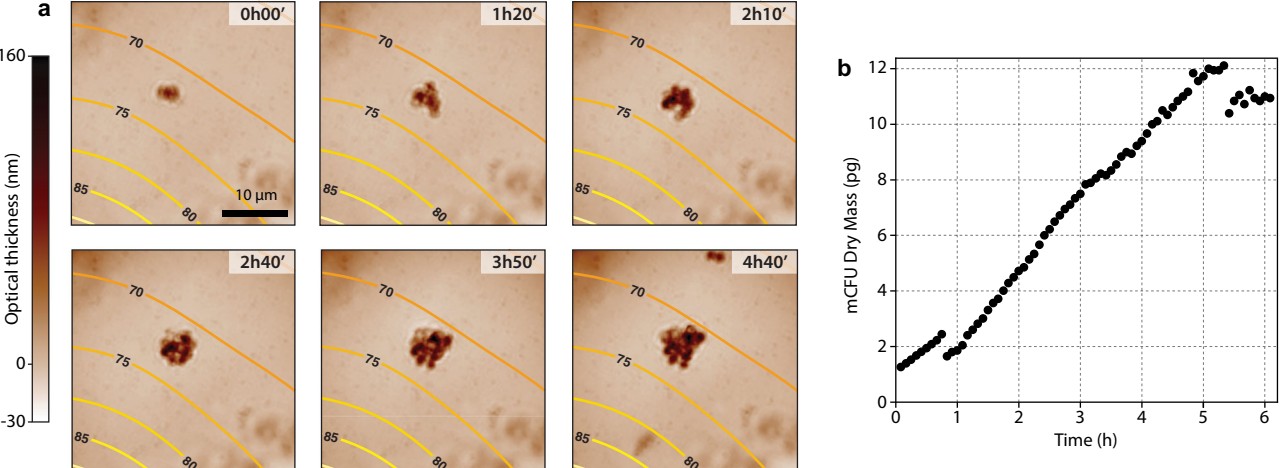

**Fig. 7 | Growth of *S. shibatae* archaea followed by CGM, and activated by laser-heating. a** Time lapse composed of 6 OT images (immersion oil, 60x, 1.25 NA objective lens), along with (**b**) the evolution of the biomass of the microscale CFU measured by CGM. See Movie M7 in Supplementary Information.

purchased from Meadowlark Optics. The holograms were calculated using the Gerchberg Saxton algorithm, as described in Ref. 39.

### Wavefront microscopy for temperature imaging and dry mass measurements

Cross-grating wavefront microscopy (CGM) is an optical microscopy technique based on the association of a 2-dimensional diffraction grating (aka cross-grating) placed at a millimetric distance from the sensor of a regular camera[33]. The most common instance of CGM, which we use in this study, is called quadriwave lateral shearing interferometry (QLSI), where the cross-grating consists of a checkerboard intensity/phase pattern, as introduced and patented by Primot et al. in 2000[34]. The vertical and horizontal lines of the grating produce a grid-like shadow on the sensor, the distortion of which can be numerically processed in real time to retrieve the optical wavefront distortion (or equivalently the phase profile) of the incoming light. When implemented on a microscope, a CGM camera can map the optical path difference, also called optical thickness (OT), of the imaged objects with a sensitivity of a faction of a nanometer[36]. In any CGM measurement, to get rid of any imperfection of the optical components or of the beam, a primary reference OT image has to be taken and subtracted to any subsequent image acquisition.

### Temperature microscopy

Temperature microscopy was achieved using the CGM camera, as described in Ref. 32. Briefly, the heating of the liquid modifies its refractive index, which gives rise to a thermal lens effect distorting an incoming light beam. This wavefront distortion is measured by CGM, and processed using a deconvolution algorithm to retrieve the temperature distribution in the liquid medium, in 3 dimensions. Temperature mapping may be performed on an area free from bacteria to obtain better image, provided the gold nanoparticles are uniformly distributed throughout the sample, which we did occasionally. The CGM reference image was acquired without heating (laser off), and subsequent images with laser on were taken at the exact same position.

### Dry mass measurements

Dry mass measurements were achieved using the same CGM camera as for temperature imaging. The CGM reference image was acquired upon rapidly moving the sample in x and y during the exposure time, as a means to average any non-uniformity in OT due to the presence of

the bacteria. From the OT images of the bacteria, their biomasses were retrieved by image integration over an area selected using a home-made Matlab segmentation algorithm (see Numerical codes subsection), and following the procedure described in Ref. 48. Briefly, we used the relation $m = \alpha^{-1} \iint \text{OT}(x,y)\mathrm{d}x\mathrm{d}y$, where $\text{OT}(x,y)$ is the optical thickness image, $m$ the dry mass and $\alpha$ a constant. We chose $\alpha = 0.18$ $\mu m^3/pg$, the typical constant for living cells.

### Sample preparation

The glass coverslip, 25 mm in diameter, 150 μm thick, covered with gold nanoparticles was deposited inside an Attofluor™ chamber (Thermofisher), gold nanoparticles face up. A pre-culture of *Geobacillus stearothermophilus* was done in LB medium overnight (200 rpm, 60 °C) prior to each day of experiments. A drop of 5 μl of *G. stearothermophilus* suspension with an optical density (OD) between 0.3 and 0.5 was deposited onto the gold nanoparticle coverslip. Then, a 18-mm circular coverslip, with a 5-mm hole in its center, was dropped on top of the drop, and an additional 5 μl of the bacterial suspension with the same OD was deposited at the center of the hole. The hole in the glass coverslip was made following our procedure described in Ref. 45 (see Supplementary Information for more information). Then, 1 ml of LB medium was added on top of the coverslip to avoid drying of the liquid layer. A last coverslip was deposited on top of the closing cap of the Attofluor™ chamber to prevent evaporation of the culture medium during incubation. For the germination experiments, we used spores that were found to occasionally cover the top coverslip after regular experiments. For the preparation of *Sulfolobus shibatae*, a similar method was used. A pre-culture of *Sulfolobus shibatae* was done in 182 medium (DSMZ) over three days (200 rpm, 75 °C).

### Fabrication of the gold nanoparticle samples

The gold nanoparticle samples were made by block copolymer micellar lithography. The procedure is detailed in Ref. 60. Briefly, micelles encapsulating gold ions are synthesized by mixing a copolymer with $HAuCl_4$ in toluene. Then, cleaned glass coverslips are dip-coated within the solution and treated by UV illumination in the presence of a reducing agent to produce gold seeds. Finally, gold seeds are grown by putting the coverslip in contact with an aqueous solution of $KAuCl_4$ and ethanolamine for 16 min, leading to a quasiperiodic and very uniform arrangement of nonspherical gold nanoparticles absorbing in the near-infrared range.

### Numerical codes

To process the interferograms into OT images, we used a home-made algorithm, detailed in Ref. 33, and provided as a Matlab package in the following public repository: https://github.com/baffou/CGMprocess. This package enables the computation of the intensity and OT images from the recorded interferograms (including the reference one) and the knowledge of the camera-grating distance.

To compute the phase patterns to be applied to the SLM to obtain a given temperature profile, we used a home-made algorithm developed previously[39,42], and that can be found in the following public repository: https://github.com/baffou/SLM_temperatureShaping. The input is the desired temperature field, that can be set either numerically, or by a monocolor bmp image.

To segment the cells and measure their dry mass, we used a Matlab algorithm that we share in the following public repository: https://github.com/baffou/CGM_magicWandSegmentation. In each image, the user has to click on the bacterium or on the mCFU of interest, adjust the sensitivity of the magic wand, and validate the selection.

### Reporting summary

Further information on research design is available in the Nature Research Reporting Summary linked to this article.

## Data availability

The data that support the finding of this study are available from the corresponding author upon reasonable request.

## Code availability

The source code used in this study are detailed in the Method section and maintained versions can be downloaded at https://github.com/baffou/, in the following repositories: SLM_temperatureShaping, CGMprocess and CGM_magicWandSegmentation.

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

## Acknowledgements

This project has received funding from the European Research Council (ERC) under the European Union's Horizon 2020 Research and Innovation Programme (grant agreement no. 772725, project HiPhore).

## Author contributions

C.M. and M.B. performed experiments and analyzed data. A.G., V.D.C., R.C. and P.F. provided their expertise in microbiology and designed cell culture protocols. L.G. contributed to sample design and fabrication. H.R. worked on the early stage of the project. G.B. conceived and

supervised the project. All the authors discussed results and contributed to writing the manuscript.

## Competing interests

The authors declare no competing interests.
