## [Peer Review File · Nature Communications]

Life at high temperature observed in vitro upon laser heating of gold nanoparticlesREVIEWER COMMENTS

Reviewer #1 (Remarks to the Author):

This manuscript reports on a new strategy to observe thermophilic micro-organisms with currently accessible microscopes. The micro-organisms are introduced into a simple microchamber built with a glass slide covered with gold nanoparticles covering the substrate. Local heating is achieved by means of a clever shaping of a laser beam, which enables the authors to deliver homogeneous heating up to high temperatures over the observation field. This new strategy has been implemented by observing the behavior of living *Geobacillus stearothermophilus* at 65°C at the single cell level. This thermophilic bacterial species was shown to grow and swim normally. Moreover, the impact of possible microscale fluid convection, cell confinement and centrifugal thermophoretic motion was carefully examined.

Although relying on technologies previously validated by the authors, this paper addresses a highly significant and demanded issue for a wide community of biologists dealing with microscopic observation of thermophilic species. In particular, the proposed strategy is definitively convincing by eliminating all the detrimental action of heating (e.g. alteration of the microscope objective), which were previously reported with alternative approaches. The work is extremely solid and the manuscript is compact. Therefore, I consider that this manuscript is relevant for publication in *Nat. Commun.*

Reviewer #2 (Remarks to the Author):

The paper by Molinaro and colleagues presents a novel platform for thermal microscopy that uses microscale heating based on thermal activation of gold nanoparticles to heat very small volumes within a laser focal plane within a prepared slide. The features of the system are described and then demonstrated using cultures of the thermophile *Geobacillus stearothermophilus*.

Since I am aware of commercially available and custom heated stages using capillaries, I was originally skeptical. However, the paper presents some good arguments for this particular system. In particular, the manufacture and operation of this system seems to be favorable, and the simultaneous measurement of biomass within the focal plane is very convenient.

Although I'm supportive, I do have some critical comments below.

1. The paper presents good arguments for the utility of the current system relative to capillary systems; however, the system is dependent on lack of cell movement (e.g., through adhering to the slides) due to the very small regions of heating. The paper makes a good case for advantages to this system (e.g., growth rate measurement at several temperatures within one field of view). However, the flipside of this is that cells that don't adhere or measurements of motility, similar to those of Wirth and colleagues, might be difficult to achieve. I would like the authors to think about and discuss these problems or present possible solutions.

2. *Geobacillus* is a very hardy moderate thermophile and cheap systems are readily available for similar experiments at similar temperatures. However, heated stages covering the temperature ranges for hyperthermophiles (e.g., 80 C and above) are not commercially available, as far as I know. As such, it was a little disappointing to see these experiments done at 65 C with *Geobacillus* rather than at 90 C, for example, with a hyperthermophile. With the system in place, it seemingly wouldn't be that difficult to add a quick experiment to demonstrate the system at higher temperatures and add that as a figure or figure panel to the existing manuscript. Line 136 mentions a hole in the coverslip. Doesn't the hole in the coverslip promote rapid dehydration of the slide? Is this one of the reasons the experiments weren't done at a higher temperature? I would like the authors to discuss the options better here and what would be optimal for hyperthermophilic conditions.

3. A lot of minor language editing is needed. I did some work on this (see below) but some editing help would ease the burden on reviewers and editors and help ensure a high-quality paper.

Minor points.

- line 13 and elsewhere: Life should not be capitalized
- line 14 "hardly achievable" should be changed to "difficult to achieve"
- line 19 remove increases (temperatures)
- line 24 and elsewhere: "a motile thermophilic bacterium"
- line 31 remove "as"
- line 32 (and similar corrections elsewhere): heat-loving microorganisms...have been observed to grow. In general, I suggest reviewing rules for use of hyphens for compound adjectives (e.g., <https://www.grammarly.com/blog/hyphen/>)
- line 33 good evidence exists for microbial growth to at least 122 C (<https://www.pnas.org/content/105/31/10949>)
- line 34 deep-sea (compound adjective)
- line 41 remove "test"
- line 42 thermophiles.
- line 45 "Microscopic observation of live cells at high temperatures is not possible using standard..."
- line 64 "heat-sealed" (compound adjective)
- line 77 "air-objectives" Please rewrite. I'm not sure what to suggest here but there is no simple term in English.
- line 98 "a motile thermophilic bacterium"
- I am now making less grammatical corrections but the authors should seek help with this
- line 120 please provide a citation or two
- line 125 "to study microorganisms"
- line 129 "mingling of the bacteria" I'm not sure what this means. Please rewrite.
- line 134 the case here is made that growing aerobes is difficult but this should be broadened to any low-aqueous solubility gas. This is a much bigger problem than just oxygen.
- line 144 and elsewhere there are extraneous references to many panels of Figure 2 that don't exist. I think they are all referring to panels of Figure 1.
- line 185 "evolution of the dry biomass" here and elsewhere should be something like "increase in dry cell biomass"
- line 96 refers to "saturation (logarithmic)" but I'm not sure why there is reference to "logarithmic". I also suggest the authors to refer to stationary phase here, which is the well-known microbiology term for saturation of growth.
- Figure 4 legend "Microbial growth at different temperatures"
- line 211 to 216: I don't think this long discussion is needed. The text here is awkward. Besides, the video is very convincing that this is motility.

Reviewer #3 (Remarks to the Author):

In this manuscript, Molinaro et al. report the adaptation of a laser-assisted heating technique for use in live-cell imaging of thermophiles. They validate their setup by successfully imaging cell division and cell motility in live *Geobacillus stearothermophilus* bacteria at 65°C. The method offers alternative solutions to existing challenges in other high-temperature microscopy techniques, such as evaporation of the culture media or insufficient speed in the control of the temperature. The most significant improvement compared to other techniques is the possibility to directly and precisely measure the local temperature distribution through quantitative phase microscopy.

The authors do a great job making sure the cells are maintained in conditions as physiological as possible – successfully so. The experiments are conducted rigorously, the text is well-written, and the figures are well-presented.

Overall, the paper is interesting, and I think the technique presented here is likely to be used in

the field. However, some points should be addressed for this study to be fully convincing.

Major comments:

1) Fig. 4: The only novel biological data in the manuscript is the direct quantification of cell growth rates and generation times through the measurement of the dry biomass, which yields values that are in accordance with what had been published before, although with greater precision.

I wish the authors had used their technique to test a hypothesis, or to answer more biologically important questions in their model organism, or another organism, that critically depend on the ability to image live cells.

2) Fig. 4: The plotted values represent measurements made in one single movie. If this is representative of multiple replicates, the authors should clarify how many biological replicates were performed.

Ideally, values averaged over three independent experiments would be the most convincing. However, if the experiment is too challenging, the authors should at least do it one more time and show that they obtain comparable results.

Minor comments:

1) Line 238: The authors state "The LA-HTM enables the use of any objective lens, including oil immersion or not, which will remain at room temperature, even with extreme temperatures reached over the field of view".

The authors should demonstrate that this is true. (All the experiments presented here were performed using a 60X 0.7 NA air objective)

Besides, the use of a high NA immersion objective would actually be preferred here over a 60X air, in the sense that it would allow the direct quantification of cell growth versus time, and the duration of cell division, in individual cells. It would also yield greater details about cell shape and morphology, or about the location of the division plane, which are critical parameters only accessible through high-resolution live-cell imaging.

2) Even though laser-assisted heating can avoid heating up the imaging environment, the authors should discuss safety issues related to the use of an IR laser (accidental eye exposure) and preventing any IR light from coming out of the oculars.

In summary, I think the authors would make a much stronger case that their technique will be broadly useful to the study of extremophiles by 1) showing that LA-HTM can be used to answer important biological questions in a species of interest, and/or 2) showing that LA-HTM is robust and can yield reproducible data, and/or 3) showing that LA-HTM can routinely be used to achieve superior spatial resolution (i.e. 100X 1.4 NA immersion).

Arthur Charles-Orszag, PhD

Reviewer #1

This manuscript reports on a new strategy to observe thermophilic micro-organisms with currently accessible microscopes. The micro-organisms are introduced into a simple microchamber built with a glass slide covered with gold nanoparticles covering the substrate. Local heating is achieved by means of a clever shaping of a laser beam, which enables the authors to deliver homogeneous heating up to high temperatures over the observation field. This new strategy has been implemented by observing the behavior of living *Geobacillus stearothermophilus* at 65°C at the single cell level. This thermophilic bacterial species was shown to grow and swim normally. Moreover, the impact of possible microscale fluid convection, cell confinement and centrifugal thermophoretic motion was carefully examined.

Although relying on technologies previously validated by the authors, this paper addresses a highly significant and demanded issue for a wide community of biologists dealing with microscopic observation of thermophilic species. In particular, the proposed strategy is definitively convincing by eliminating all the detrimental action of heating (e.g. alteration of the microscope objective), which were previously reported with alternative approaches. The work is extremely solid and the manuscript is compact. Therefore, I consider that this manuscript is relevant for publication in *Nat. Commun.*

We thank Reviewer 1 for her/his positive feedback.

Reviewer #2

The paper by Molinaro and colleagues presents a novel platform for thermal microscopy that uses microscale heating based on thermal activation of gold nanoparticles to heat very small volumes within a laser focal plane within a prepared slide. The features of the system are described and then demonstrated using cultures of the thermophile *Geobacillus stearothermophilus*.

Since I am aware of commercially available and custom heated stages using capillaries, I was originally skeptical. However, the paper presents some good arguments for this particular system. In particular, the manufacture and operation of this system seems to be favorable, and the simultaneous measurement of biomass within the focal plane is very convenient.

Although I'm supportive, I do have some critical comments below.

1. The paper presents good arguments for the utility of the current system relative to capillary systems; however, the system is dependent on lack of cell movement (e.g., through adhering to the slides) due to the very small regions of heating. The paper makes a good case for advantages to this system (e.g., growth rate measurement at several temperatures within one field of view). However, the flipside of this is that cells that don't adhere or measurements of motility, similar to those of Wirth and colleagues, might be difficult to achieve. I would like the authors to think about and discuss these problems or present possible solutions.

The reviewer points out here the main limitation of our approach, which is indeed an important point. We now dedicate a new paragraph to further explain this limitation (L302 to 307):

The main limitation of the laser-heating approach we report in this study is that cells that don't adhere, or that swim, may move away from the field of view and be difficult to investigate. A workaround could be to use low-magnification objectives to achieve a much more extended temperature increase, over a few 100s of microns. This caution would come along with a reduced spatial resolution but if the aim is to study the motion of micro-organisms, a high spatial resolution is not a requirement.

2. *Geobacillus* is a very hardy moderate thermophile and cheap systems are readily available for similar experiments at similar temperatures. However, heated stages covering the temperature ranges for hyperthermophiles (e.g., 80 C and above) are not commercially available, as far as I know. As such, it was a little disappointing to see these experiments done at 65 C with *Geobacillus* rather than at 90 C, for example, with a hyperthermophile. With the system in place, it seemingly wouldn't be that difficult to add a quick experiment to demonstrate the system at higher temperatures and add that as a figure or figure panel to the existing manuscript. Line 136 mentions a hole in the coverslip. Doesn't the hole in the coverslip promote rapid dehydration of the slide? Is this one of the reasons the experiments weren't done at a higher temperature? I would like the authors

to discuss the options better here and what would be optimal for hyperthermophilic conditions.

Dehydration is avoided, at least over 2 or 3 days, thanks to the large amount of medium we add on top of the drilled coverslip (see Methods part). However, we did not explain that this evaporation is further prevented by the top coverslip we put on the metallic (although the top coverslip was represented in Figure 1b). We now write:

A top coverslip is put on top of the metallic dish (Figure 1b) to avoid evaporation of the medium and enable observation over at least a couple of days. Because this top coverslip is not sealed, additional medium can be easily added at any time, if need be.

There is no identified limit in the temperature that can be reached. In the original version we did not discuss much this point, because we wanted to keep it for a future publication, but we agree it would be interesting to discuss it here, as it is a natural question. And we would not like the reader to think there is a limitation in the temperature range while there is not.

In principle, the approach we propose does not only enable to reach 90°C, but also temperatures higher than 100°C without boiling. We can reach a superheated state of water by heating layers of nanoparticles as demonstrated in a publication of ours, in 2014:¹ This phenomenon comes from the absence of nucleation points, which prevents boiling until ~200°C. We already demonstrated an application of this phenomenon, in hydrothermal chemistry, in 2016.²

In the revised version, here is what we have done to go along with what proposes the referee:

- We present additional studies with another cell type, the hyperthermophilic archaeon *Sulfolobus Shibatae* (not a bacterium), living around 80°C (no longer 65°C).
- We tell that there is a priori no limit in the temperature range that can be achieved, based on our previous publications, in a new section of the Discussion part:

The temperature range the LA-HTM can potentially reach. The study of *S. shibatae*, optimally living at 80°C, shows that the LA-HTM can be applied to the study of hyperthermophiles, not only to simple thermophiles. In principle, there exists no limit in the temperature range that can be achieved using the LA-HTM and temperatures higher than 100°C could even be achieved at ambient pressure without boiling, as demonstrated by our group³⁸ for applications in hydrothermal chemistry at ambient pressure using laser heating of gold nanoparticles.⁴⁰ Thus, the LA-HTM could potentially be used to observe unprecedented hyperthermophiles with standard high-resolution optical microscopes, in standard conditions (i.e. ambient pressure).

¹ Super-Heating and Micro-Bubble Generation around Plasmonic Nanoparticles under cw Illumination, G. Baffou, J. Polleux, H. Rigneault, S. Monneret, Journal Physical Chemistry C 118, 4890 (2014)

² Light-Assisted Solvothermal Chemistry Using Plasmonic Nanoparticles, H. M. L. Robert, F. Kundrat, E. Bermudez-Urena, H. Rigneault, S. Monneret, R. Quidant, J. Polleux, G. Baffou, ACS Omega 1, 2-8 (2016)

3. A lot of minor language editing is needed. I did some work on this (see below) but some editing help would ease the burden on reviewers and editors and help ensure a high-quality paper.

Minor points.

- line 13 and elsewhere: Life should not be capitalized
- line 14 “hardly achievable” should be changed to “difficult to achieve”
- line 19 remove increases (temperatures)
- line 24 and elsewhere: “a motile thermophilic bacterium”
- line 31 remove “as”
- line 32 (and similar corrections elsewhere): heat-loving microorganisms...have been observed to grow. In general, I suggest reviewing rules for use of hyphens for compound adjectives (e.g., <https://www.grammarly.com/blog/hyphen/>)
- line 33 good evidence exists for microbial growth to at least 122 C (<https://www.pnas.org/content/105/31/10949>)
- line 34 deep-sea (compound adjective)
- line 41 remove “test”
- line 42 thermophiles.
- line 45 “Microscopic observation of live cells at high temperatures is not possible using standard...”
- line 64 “heat-sealed” (compound adjective)
- line 77 “air-objectives” Please rewrite. I’m not sure what to suggest here but there is no simple term in English.
- line 98 “a motile thermophilic bacterium”
- I am now making less grammatical corrections but the authors should seek help with this
- line 120 please provide a citation or two
- line 125 “to study microorganisms”
- line 129 “mingling of the bacteria” I’m not sure what this means. Please rewrite.
- line 134 the case here is made that growing aerobes is difficult but this should be broadened to any low-aqueous solubility gas. This is a much bigger problem than just oxygen.
- line 144 and elsewhere there are extraneous references to many panels of Figure 2 that don’t exist. I think they are all referring to panels of Figure 1.
- line 185 “evolution of the dry biomass” here and elsewhere should be something like “increase in dry cell biomass”
- line 96 refers to “saturation (logarithmic)” but I’m not sure why there is reference to “logarithmic”. I also suggest the authors to refer to stationary phase here, which is the well-known microbiology term for saturation of growth.
- Figure 4 legend “Microbial growth at different temperatures”
- line 211 to 216: I don’t think this long discussion is needed. The text here is awkward. Besides, the video is very convincing that this is motility.

We warmly thank the reviewer for her/his help on the writing. All these corrections have been implemented one by one, without exception.

Reviewer #3

In this manuscript, Molinaro et al. report the adaptation of a laser-assisted heating technique for use in live-cell imaging of thermophiles. They validate their setup by successfully imaging cell division and cell motility in live *Geobacillus stearothermophilus* bacteria at 65°C. The method offers alternative solutions to existing challenges in other high-temperature microscopy techniques, such as evaporation of the culture media or insufficient speed in the control of the temperature. The most significant improvement compared to other techniques is the possibility to directly and precisely measure the local temperature distribution through quantitative phase microscopy.

The authors do a great job making sure the cells are maintained in conditions as physiological as possible – successfully so. The experiments are conducted rigorously, the text is well-written, and the figures are well-presented.

Overall, the paper is interesting, and I think the technique presented here is likely to be used in the field. However, some points should be addressed for this study to be fully convincing.

Major comments:

1) Fig. 4: The only novel biological data in the manuscript is the direct quantification of cell growth rates and generation times through the measurement of the dry biomass, which yields values that are in accordance with what had been published before, although with greater precision.

I wish the authors had used their technique to test a hypothesis, or to answer more biologically important questions in their model organism, or another organism, that critically depend on the ability to image live cells.

To go along with what the referee is suggesting, and to stick to the bacteria we have been investigating, we decided to focus on a problem of high interest for the community, although poorly investigated: the mechanism of sporulation/germination of *G. stearothermophilus*. Albeit well documented, no sporulation/germination events of *G. stearothermophilus* have ever been observed in real time. We report now the first observation of a germination event of *G. stearothermophilus*. We could measure the initial growth rate and evidence that the re-growth is stochastic.

Moreover, our new study of *Sulfolobus shibatae*, following the suggestion of Reviewer 1 to study organisms living at higher temperatures, led us to the observation that this microorganism does not follow an exponential growth, but rather a linear growth, providing some more substance to the debate of linear/exponential growth of micro-organisms.

2) Fig. 4: The plotted values represent measurements made in one single movie. If this is representative of multiple replicates, the authors should clarify how many biological replicates were performed. Ideally, values averaged over three independent experiments

would be the most convincing. However, if the experiment is too challenging, the authors should at least do it one more time and show that they obtain comparable results.

We conducted another set of measurements in that sense. However, we would not like to average several measurements because we would like not to convey the message that several experiments must be conducted to plot this curve. So we preferred to plot the several measurements on a same graph, instead of averaging them. Moreover, averaging would not be easy to do anyway because we would need measurements at the same temperature values, but we don't decide in which isotherms the microscale CFU decide to show on the field of view.

To avoid overloading Figure 4, and because Figure 4 is related to Figures 2 and 3, we did not touch Figure 4, but added a new section in the article and a new figure in Suppl. Info. about the reproducibility. In Figures S8 & S9, we plot two sets of measurements and provide discussion on the reproducibility.

Figure S8: (a) Microbial growth at different temperatures. Data acquired from a single movie of bacteria growing within a temperature gradient (movie M4). (b) Same as (a) in a semi-log scale. (c) Growth rate τ and generation time calculated from the linear fits of (b).

Figure S9: Comparison of the two growth rate functions plotted in Figs. 4c and S8c.

Minor comments:

1) Line 238: The authors state “The LA-HTM enables the use of any objective lens, including oil immersion or not, which will remain at room temperature, even with extreme temperatures reached over the field of view”.

The authors should demonstrate that this is true. (All the experiments presented here were performed using a 60X 0.7 NA air objective)

Besides, the use of a high NA immersion objective would actually be preferred here over a 60X air, in the sense that it would allow the direct quantification of cell growth versus

time, and the duration of cell division, in individual cells. It would also yield greater details about cell shape and morphology, or about the location of the division plane, which are critical parameters only accessible through high-resolution live-cell imaging.

This was indeed not consistent. All the new studies (Figures 6, 7, S7 and Movies M4, M6, M7) that we now show in the revised version of the manuscript have been done with high-NA immersion oil objective (60x, 1.25 NA).

2) Even though laser-assisted heating can avoid heating up the imaging environment, the authors should discuss safety issues related to the use of an IR laser (accidental eye exposure) and preventing any IR light from coming out of the oculars.

We added this paragraph in the revised version:

The only new safety issue stems from the presence of a high-power laser beam (up to typically 100 mW) within the setup and possibly throughout the ocular, which requires the wearing of safety glasses.

In summary, I think the authors would make a much stronger case that their technique will be broadly useful to the study of extremophiles by 1) showing that LA-HTM can be used to answer important biological questions in a species of interest, and/or 2) showing that LA-HTM is robust and can yield reproducible data, and/or 3) showing that LA-HTM can routinely be used to achieve superior spatial resolution (i.e. 100X 1.4 NA immersion).

Arthur Charles-Orszag, PhD

Dear Pr. Charles-Orszag, thank your valuable comments. As detailed above, we did our best to fulfil these three recommendations.

REVIEWERS' COMMENTS

Reviewer #2 (Remarks to the Author):

The reviewers have satisfied all of my concerns from the original review. Well done!
- Brian Hedlund, Reviewer 2 from the first round

Reviewer #3 (Remarks to the Author):

Review of the revised version of Molinaro, Bénéfice, et al., 2021, "Life at high temperature observed in vitro upon laser heating of gold nanoparticles" submitted to Nature Communications

Date: 06/22/2022

In the revised version of their manuscript, Molinaro, Bénéfice, and colleagues convincingly demonstrate that LA-HTM 1) is suitable for use with high-NA objectives, 2) yields reproducible and robust biological measurements, and 3) is a versatile tool that can be used to answer important biological questions in the field of extreme microbiology.

Of great interest are the newly added first-time imaging of cell division at the single cell level in the hyperthermophilic crenarchaeon *S. shibatae*, and the even more impressive first-time imaging of spore germination in the thermophilic bacterium *G. stearothermophilus*.

In summary, the presented work is scientifically sound, robust, well-written, and addresses key questions in the field of extreme microbiology. There is no doubt it will greatly benefit our community, and I recommend it be published in Nature Communications.

Arthur Charles-Orszag, PhD